# Kinematic Comparison between Medially Congruent and Posterior-Stabilized Third-Generation TKA Designs

**DOI:** 10.3390/jfmk6010027

**Published:** 2021-03-15

**Authors:** Stefano Ghirardelli, Jessica L. Asay, Erika A. Leonardi, Tommaso Amoroso, Thomas P. Andriacchi, Pier Francesco Indelli

**Affiliations:** Department of Orthopaedic Surgery, Stanford University School of Medicine, Stanford, The Palo Alto Veterans Affairs Health Care System (PAVAHCS), Palo Alto, CA 94304, USA; ghirardelli.stefano@gmail.com (S.G.); jdemarre@stanford.edu (J.L.A.); erileo@stanford.edu (E.A.L.); tommy.amo@hotmail.it (T.A.); tandriac@stanford.edu (T.P.A.)

**Keywords:** TKA, gait, total knee arthroplasty, medial pivot, posterior-stabilized, MC, PS, knee, kinematic, biomechanics

## Abstract

**Background**: This study compares knee kinematics in two groups of patients who have undergone primary total knee arthroplasty (TKA) using two different modern designs: medially congruent (MC) and posterior-stabilized (PS). The aim of the study is to demonstrate only minimal differences between the groups. **Methods**: Ten TKA patients (4 PS, 6 MC) with successful clinical outcomes were evaluated through 3D knee kinematics analysis performed using a multicamera optoelectronic system and a force platform. Extracted kinematic data included knee flexion angle at heel-strike (KFH), peak midstance knee flexion angle (MSKFA), maximum and minimum knee adduction angle (KAA), and knee rotational angle at heel-strike. Data were compared with a group of healthy controls. **Results:** There were no differences in preferred walking speed between MC and PS groups, but we found consistent differences in knee function. At heel-strike, the knee tended to be more flexed in the PS group compared to the MC group; the MSKFA tended to be higher in the PS group compared to the MC group. There was a significant fluctuation in KAA during the swing phase in the PS group compared to the MC group, PS patients showed a higher peak knee flexion moment compared to MC patients, and the PS group had significantly less peak internal rotation moments than the MC group. **Conclusions:** Modern, third-generation TKA designs failed to reproduce normal knee kinematics. MC knees tended to reproduce a more natural kinematic pattern at heel-strike and during axial rotation, while PS knees showed better kinematics during mid-flexion.

## 1. Background

Total knee arthroplasty (TKA) is considered one of the most effective surgical procedures for the treatment of advanced and degenerative tricompartmental osteoarthritis (OA) of the knee [1,2].

Unfortunately, only 80% of TKA patients are completely satisfied, leaving 20% of them with serious limitations in their activities of daily living (ADLs) [3] when compared with their age-matched peers [4] despite design evolution and modern adaptive technology. This is more evident in young patients asking to return to sports and other high recreational activities after surgery [5]. Many patients still complain of subjective micro- or macro-instability [6], altered proprioception, and abnormal knee joint awareness: their dissatisfaction is confirmed by poor range of motion (ROM), chronic effusions, inability to use the stairs in a reciprocating way, inability to kneel or squat [7,8] and occasional antalgic gait.

In the last 40 years, TKA designs have been radically improved to solve the problems found with earlier designs [9,10]. Recently, third-generation TKA systems have been designed to restore normal knee kinematics, thanks to updated geometry incorporating asymmetric tibiofemoral articulations and an increment in implant modularity [11]. In general, the rationale for these more modern designs is to provide more natural joint proprioception compared to traditional, less conforming articular surface designs.

Native and post-TKA knee kinematics differ significantly: standard and dynamic fluoroscopy [12,13], Roentgen sterephotogrammetric analysis [14], gait analysis [15], and in vitro techniques [16] have shown the major differences between normal and TKA knees, theoretically justifying the still very-high dissatisfaction rate among TKA patients. 

In the last 10 years, normal knee kinematics during ambulatory activities have been studied, thanks to improvement in technology. Many reports, including one from the senior author of the current study [17], support the evidence that a big difference in kinematic behavior is present when knees are tested in the stance phase of gait (having the joint center of rotation on the lateral side) or during swing phases of gait, stair ascent activities, and squatting (having a medial center of rotation) [18,19]. The reproduction of these dual pivoting kinematics is extremely challenging since the anterior cruciate ligament (ACL), which is routinely removed during TKA, plays a major role in normal knee mechanics [20]. The current authors have previously described a surgical technique [21], adapted to a third-generation TKA medial pivot design [22,23], that has shown very satisfactory results when compared to other classical techniques and posterior-stabilized implants [11]. 

The aim of this study is to compare parameters of kinematics during normal gait parameters (knee flexion–extension angle, adduction–abduction angle, internal–external tibial rotation, peak knee flexion moment, first peak knee adduction moment, and peak knee internal rotation moment) of TKA patients having either a modern third-generation posterior-stabilized TKA design or a medially-congruent (but not fully congruent, or “ball in socket”) design from the same manufacturer and to compare kinetics and kinematics of people with a TKA to a matched group of healthy controls. The hypothesis is that the kinematic parameters between medially congruent and PS TKA designs significantly differ from healthy controls when evaluated during standard gait analysis. The authors also hypothesize to demonstrate only minimal differences between the two prosthetic implants. 

## 2. Materials and Methods

This was a retrospective case–control study: patients who underwent primary TKA at the authors’ institution because of severe monolateral knee osteoarthritis were included in this study. All patients gave their informed consent before their inclusion in the study after receiving IRB approval (Stanford University IRB 42630, 19 March 2018). Preoperative initial inclusion criteria included age greater than 40 years and radiographically diagnosed monoliteral, tricompartmental OA. Preoperative exclusion criteria included OA of the hips, ankles, or contralateral knee, presence of chronic inflammatory diseases, age greater than 85 years, body mass index (BMI) greater than 35 kg/m^2^, varus deformity of more than 15°, and total knee or hip replacement in either limb. Patient demographics are described in Table 1. 

All patients received a third-generation TKA design (Persona, Zimmer-Biomet, Warsaw, IN, USA) performed by a single surgeon (PFI) who had more than five years of clinical experience with the system at the beginning of the current study [21,22,23]. This TKA system has two options for the J-curve femoral design (posterior-stabilized (PS) and cruciate-retaining (CR)) and a single anatomical tibial baseplate that allows us to use two different polyethylene inserts: PS (for the PS femur) and medially congruent (MC; when a CR femur is selected). The PS polyethylene design insert has symmetric tibiofemoral congruency on the medial and lateral compartments; in contrast, the MC insert is fully congruent medially (1:1 radius) with respect to the femoral condyle, and it has a dwell point that is 1.5 mm more posterior than the CR and PS inserts (Figure 1). An identical, previously described, surgical technique was used in all cases [21,22,23]: this was a combination of gap-balancing in extension and measured resection in flexion, with the constant removal of the posterior–cruciate ligament (PCL). All patients followed an identical standard postoperative rehabilitation protocol, including weight-bearing, as tolerated with crutches on postoperative day 1; clinical improvements were assessed preoperatively and at 3-month, 6-month, and final follow-up (FU).

Ten TKA (4 posterior-stabilized PS and 6 medially congruent MC) were matched at a minimum of 9 months FU by gender, age, BMI, and operating surgeon to 10 healthy controls. Postoperative inclusion criteria for both TKA groups were a contralateral knee not being replaced, obtaining a high score according to the Western Ontario and McMaster Universities Arthritis Index (WOMAC) average score [20], Knee Injury and Osteoarthritis Outcome Score (KOOS) average pain score [24], and Forgotten Joint Score [25]. Patients in both groups were matched by age (TKAs: 65.8 ± 8.8 years; control group: 59.4 ± 7.9 years), sex (all males), and BMI (TKAs: 31.5 ± 4.9 kg/m^2^; control group: 30.3 ± 4.6 kg/m^2^) (Table 1). All patients had a full extension of the knee and at least 125° of active flexion during examination prior to the test. 

### Gait Analysis

Postoperative (9-month minimum FU) 3D knee kinematic analysis, performed using a multicamera optoelectronic system (Qualisys AB, Gothenburg, Sweden) and a force platform (Bertec Corporation, Columbus, OH, USA) embedded in the middle of a 10-m walkway, was compared between PS TKA patients, MC TKA patients, and healthy controls. Camera and force data were synchronized and collected at 120 Hz. Marker data were collected using the previously described point cluster.

A previously described technique (PCT) [26] was used to keep track of the relative motion of the lower extremity during testing: the anatomical landmarks were first determined through palpation, and 15 markers (9 on the thigh and 6 on the leg) were secondarily placed on the skin. Data gathering included performing a static trial in order to obtain anatomical reference frames, using inverse dynamics to calculate knee joint angles and moments and, finally, determining the different knee joint moments during multiple phases of gait (BioMove software, Stanford University, Stanford, CA, USA) [27].

Subjects performed three walking trials at their self-selected normal pace. Extracted kinematic data included knee flexion angle at heel-strike (KFH), peak midstance knee flexion angle (MSKFA), maximum and minimum knee adduction angle (KAA), and knee rotational angle at heel-strike. Peak joint moments of the knee included first peak knee adduction (KAM), peak knee flexion moment (KFM), and peak internal knee rotational moment (KIRM). The external moments were calculated using a standard inverse dynamic approach and were normalized to percent body weight and height (%BW*Ht) to allow comparison between individuals. Data were averaged for the walking trials. Clinical scores (KOOS and Forgotten Joint Score) were also collected at the time of the gait test. Differences between PS TKA and MC TKA were determined through standard Student’s *t*-tests, and the differences between the TKA group as a whole and the control group were also determined through standard Student’s *t*-tests. Significance was set at *p* < 0.05, with trends of *p* < 0.15.

## 3. Results

Patient demographics are described in Table 1. There was no significant difference in BMI, sex, age, or clinical score between TKA patients and healthy controls. The study population was divided into three groups: MC TKA, PS TKA, and healthy controls. There were no differences in preferred walking speed between MC and PS TKA groups (1.25 m/s and 1.30, respectively (*p* = 0.69)), but there were consistent differences in knee function between the groups. 

### 3.1. Knee Flexion Angle at Heel-Strike (KFH)

During the heel-strike portion of the gait, the knee tended to be more flexed in the PS group compared to the MC group (MC = 3.07°, PS = 7.95°; *p* = 0.059; Table 2, Figure 2). This is mainly due to more forward inclination of the shank and less forward inclination of the thigh, as previously demonstrated by the senior author [27]. We did not find any significant correlations between KFH and patient-reported outcomes (PROs). 

### 3.2. Midstance Knee Flexion Angle

There was a trend that the average midstance knee flexion angle in the PS TKA group (19.2°) was higher than the MC TKA group (14°; *p* = 0.13; Table 2, Figure 2). The degree of the midstance knee flexion angle correlated to pain, according to the KOOS pain score in the two groups. Patients with higher midstance knee flexion angles reported better pain outcomes (*p* = 0.02; Figure 3). 

### 3.3. Knee Adduction Angle (KAA) and Knee Adduction Moment (KAM)

There were significant fluctuations in the knee adduction angle during the swing phase in the PS TKA group compared to the MC TKA group (Table 2, *p* = 0.02, Figure 4); there was no correlation between KAA variability and PROs. Interestingly, the TKR groups tended to have less knee adduction angle excursion (peak-to-peak) compared to healthy controls (Figure 4b). The first peak knee adduction moment (KAM1) demonstrated no significant differences between the groups, and no correlations to PROs were found. 

### 3.4. Peak Knee Flexion Moment (KFM)

There was a trend that PS patients showed higher peak KFM (3.78%BW*Ht) than the MC group (2.34%BW*Ht) at 25% of the gait cycle (Table 2, *p* = 0.12, Figure 5a). There was also a trending correlation between peak KFM and the KOOS pain score; patients reporting better KOOS scores had higher peak KFM (*p* = 0.07; Figure 5b). 

### 3.5. Knee Rotational Moment (KRM)

The knee internal rotation moment reached its peak during the late stance phase in all groups. Interestingly, the PS TKA group had significantly less peak internal rotation during stance (*p* = 0.02) when compared to the MC TKA group. We found no correlation between KRM and PROs. 

### 3.6. TKA as a Whole Compared to the Healthy Control Group

Both TKA groups had significantly less rotation at heel-strike compared to controls (*p* = 0.04), fewer late stance peak extension moments compared to controls (*p* = 0.02), and greater peak knee adduction angles during the swing phase than controls (*p* = 0.04; Table 1).

## 4. Discussion

This study shows that even modern, third-generation TKA systems can fail to reproduce normal knee kinematics. The lack of full knee extension during stance, the absence of the “screw-home mechanism”, typical of an ACL functioning knee, and increased mediolateral instability (>KAA) during the swing phase still represent major differences, from a proprioceptive and muscular recruitment point of view, between normal and prosthetic knees. No major differences were demonstrated in terms of kinematic parameters between MC and PS TKA designs: if MC knees tended to reproduce a more natural kinematic pattern at heel-strike, PS knees showed better KFM and better quadricep recruitment during mid-flexion. 

The current authors historically favored pivoting types of TKA designs because of strong evidence from registries, single-center clinical studies, gait laboratories, and patient surveys that those designs provide outcomes that are at least equivalent but are often superior to TKA designs characterized by a higher level of intra-articular constraint [27,28,29]. The current study’s senior author previously reported that knees showed a lateral pivot during heel-strike and early flexion gait phases [13], while a medial pivot pattern predominates in later flexion [24]. The reproduction of this “dual-pivoting” kinematics is related to significant clinical benefits and high patient satisfaction [30,31]. 

The analysis of the different gait phases in this study showed slight differences when compared to the current literature, especially in relation to the definition of the extensor mechanism/hamstrings ideal ratio during gait [32,33,34,35,36]. At heel-strike, PS knees tended to be more flexed than MC knees: this “flexion contracture” during the loading response phase of gait has been previously reported in patients following anterior cruciate ligament reconstruction and total knee arthroplasty as a strategy to limit the demands placed on their quadriceps [32,33]. We hypothesized that an intrinsic instability can be detected in PS knees, historically characterized by “paradoxical motion” in the early phase of gait [13], leading to recruitment of the hamstrings as secondary anteroposterior stabilizers. We also hypothesized that MC knees, having design-related medial intrinsic stability, might reduce the need for flexor co-contraction, possibly resulting in a functionally stronger knee in the early phase of gait [34]. 

We also confirmed that hamstrings exert a strong knee flexion moment (KFM) that counteracts the extensor torque. In our comparative series, PS knees showed higher peak KFM than MC knees: this finding is correlated with higher clinical scores. Interestingly, other authors have hypothesized that patients who utilize hamstring contraction to increase anteroposterior stability following total knee arthroplasty reduce their available extensor torque, which may have a negative effect on function [35,36]. 

The current study also demonstrates a significant difference in knee adduction angle (KAA) between MC and PS knees; a similar difference was found between TKA patients and normal, healthy controls. Interestingly, KAM, which is related to the distribution of the medial/lateral load on the knee [37], was not differentiated between the two TKA groups, suggesting that the surgical technique, more than the design itself, may play a bigger role in it. A previous model by Schipplein and Andriacchi [38] showed dynamic joint stability during walking: physiologically high KAA is present in normal and osteoarthritic knees, where the joint opens laterally and transfers the entire joint reaction through the medial compartment; co-contraction of antagonist muscle action was needed to reduce KAA and maintain stability during gait. In the current study, there was a significant “excursion” in KAA during the entire gait test in the PS group when compared to the MC group. 

Finally, this study analyzed differences in rotation moment (KRM) between TKA patients and healthy controls. Several kinematic studies demonstrated more normal tibial axial rotation when medially constrained or dual-pivoting TKA designs were used [17,18,39,40,41,42]. Our study suggests that the surgical center of rotation and the axial rotation of the knee can be modulated by lowering the tibiofemoral level of constraint [42]. In fact, MC TKAs showed significantly more peak internal rotation moments during stance than PS TKAs. As expected, TKA patients had knees that were more internally rotated at heel-strike compared to the contralateral and control group patients; this finding is typical of ACL-deficient knees [13]. 

While these results showed differences between medially constrained, posterior-stabilized, and normal knees, not being able to generalize these results represents the main limitation of the study. First, the small population and the short follow-up make it difficult to conclude that the gait change outcomes highlighted here can be applied to the general TKA population. Because of the small cohorts, strong statistical support for the results has not been achieved. 

Acknowledging these limitations, the results of this study are consistent with the experimental work of the current and other authors [11,17,20,21,22,23,24,27,34,38], where contemporary pivoting, fixed-bearing TKA designs have failed to reproduce normal knee kinematics [43]. On the other hand, the TKA pivoting design evaluated in this study and other dual-pivoting designs studied by different authors have demonstrated better-controlled kinematics that are correlated to better active flexion [23] and better functional muscle coordination [44,45], favoring clinically detectable improved knee stability. From a purely surgical perspective, after a precise surgical balancing of ligaments and soft tissues has been obtained, the medial-conforming design studied in this comparative series, characterized by a 1:1 medial compartment radius ratio and correlated to the asymmetrical tibial baseplate, gives a clear technical advantage in the correct reproduction of the medial dwell point of a natural knee. Finally, a major limitation is represented by the small number of participants: this study was conducted during the Covid-19 pandemic, and patient access to the gait analysis lab was limited by institutional (PAVAHCS and Stanford University) regulations. Because of cohort size, further large-scale studies are needed to justify the authors’ preliminary findings.

## 5. Conclusions

In conclusion, modern, third-generation pivoting TKA designs are intended to guarantee more natural knee proprioception and intrinsic stability, with the intent of producing superior clinical outcomes when compared with designs characterized by higher levels of intra-articular constraint. As more studies specifically target the comparison of modern TKA designs and better modern technology is developed to help surgeons reproduce more physiologic knee kinematics, confidence in these conclusions will eventually increase.

## Figures and Tables

**Figure 1 jfmk-06-00027-f001:**
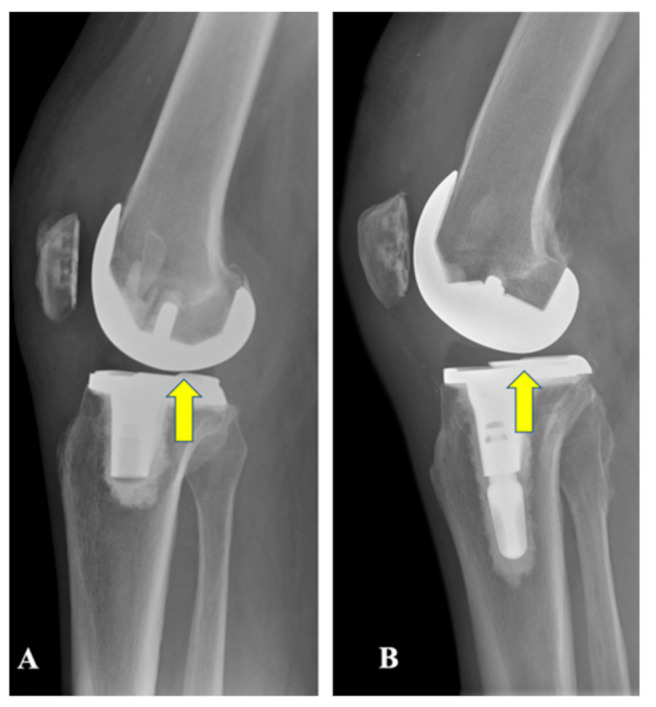
(**A**) Right medially congruent (MC) total knee arthroplasty (TKA). (**B**) Right posterior-stabilized (PS) TKA. The tibiofemoral dwell point in the MC TKA is more posterior than in the PS TKA.

**Figure 2 jfmk-06-00027-f002:**
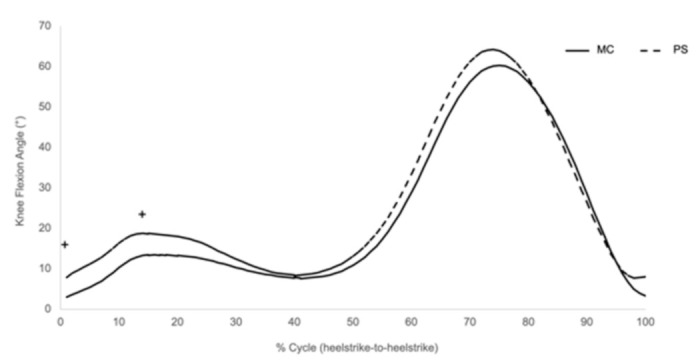
Knee flexion angle at heel-strike (KFH) and peak at midstance (MSKFA) tended to be more flexed in the PS group compared to the MC group. ^+^ Statistical trend (*p* < 0.15).

**Figure 3 jfmk-06-00027-f003:**
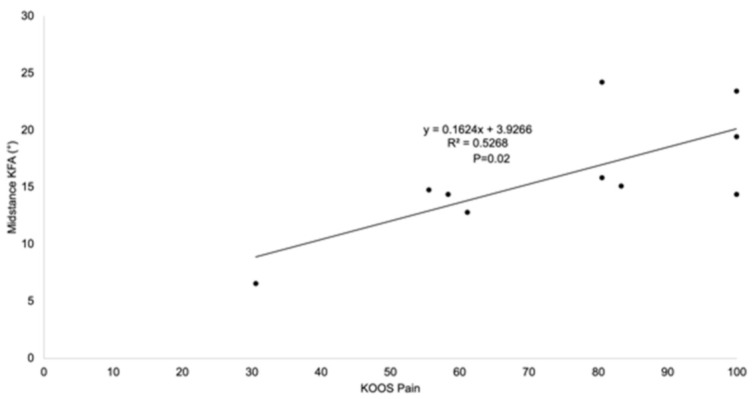
Patients with higher midstance knee flexion angles reported better KOOS pain outcomes (*p* = 0.02).

**Figure 4 jfmk-06-00027-f004:**
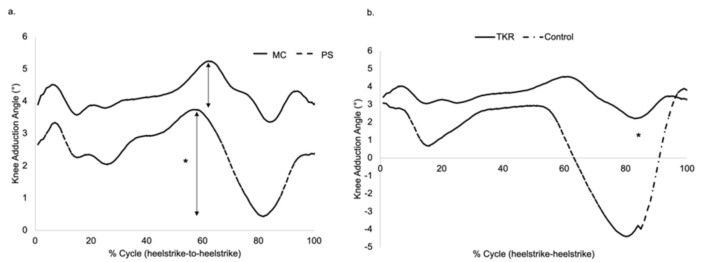
(**a**) A significant fluctuation in knee adduction angle (KAA) was demonstrated during the swing phase in the PS group compared to the MC group. (**b**) Both TKA groups showed less KAA fluctuation compared to control. * *p* < 0.05.

**Figure 5 jfmk-06-00027-f005:**
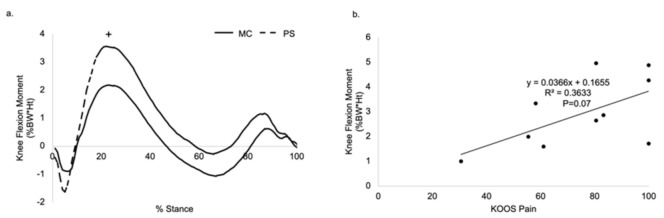
(**a**). The PS TKA group tended to have higher peak KFM than the MC TKA group. (**b**). Patients reporting better KOOS pain scores had higher peak KFM. ^+^ Statistical trend (*p* < 0.15).

**Table 1 jfmk-06-00027-t001:** Subjects Demographics at Baseline. TKA: Total Knee Arthroplasty group; BMI: body mass index; WOMAC: Western Ontario and McMaster Universities Arthritis Index; KOOS: Knee injury and Osteoarthritis Outcome Score; KFH: knee flexion angle at heel-strike; KFA: knee flexion angle; KAA: knee adduction angle; KAM: knee adduction moment; KFM: peak knee flexion moment; KIRM: knee rotational moment; N.S.: no statistically significant difference.

	TKA	Healthy Controls	*p*-Value
Age (years)	65.8 ± 8.8	59.4 ± 7.9	N.S.
Sex	10 males	10 males	N.S.
BMI (kg/m^2^)	31.5 ± 4.9	30.3 ± 4.6	N.S.
KOOS pain (points)	75.0 ± 23.1	98.9 ± 1.9	<0.01
KOOS symptoms (points)	71.4 ± 21.9	97.1 ± 4.1	<0.01
KOOS ADL (points)	81.2 ± 18.2	99.7 ± 0.9	<0.01
KOOS Sports (points)	60.0 ± 31.8	99.0 ± 2.1	<0.01
KOOS QOL (points)	63.1 ± 31.0	96.9 ± 5.3	<0.01
Forgotten Joint Score (points)	50.2 ± 38.1	N/A	N/A
KFH (°)	5.0 ± 4.1	4.5 ± 3.6	N.S.
Midstance KFA (°)	16.1 ± 5.2	20.7 ± 5.5	0.07 (trend)
Tibial rotation at heel-strike (°)	5.8 ± 5.4	11.2 ± 5.5	0.04
Peak KAA during swing (°)	0.2 ± 4.2	−5.1 ± 5.7	0.04
KAA excursion during swing (°)	6.0 ± 2.3	9.4 ± 4.9	0.07 (trend)
KAM1 (%BW*Ht)	2.16 ± 0.52	2.38 ± 0.67	N.S.
Peak KFM (%BW*Ht)	2.91 ± 1.41	3.27 ± 1.08	N.S.
Peak KIRM (%BW*Ht)	0.78 ± 0.17	0.77 ± 0.27	N.S.

**Table 2 jfmk-06-00027-t002:** Results. MC TKA: medially-congruent total knee arthroplasty; PS TKA: posterior-stabilized total knee arthroplasty; BMI: body mass index; WOMAC: Western Ontario and McMaster Universities Arthritis Index; KOOS: Knee injury and Osteoarthritis Outcome Score; KFH: knee flexion angle at heel-strike; KFA: knee flexion angle; KAA: knee adduction angle; KAM: knee adduction moment; KFM: peak knee flexion moment; KIRM: knee rotational moment; N.S.: no statistically significant difference.

	MC TKA	PS TKA	*p*-Value
Age (years)	63.8 ± 9.2	68.8 ± 4.0	N.S.
Sex	6 males	4 males	N.S.
BMI (kg/m^2^)	32.2 ± 6.1	30.5 ± 2.7	N.S.
KOOS pain (points)	75.9 ± 26.5	73.6 ± 20.8	N.S.
KOOS symptoms (points)	75.0 ± 23.6	66.1 ± 21.1	N.S.
KOOS ADL (points)	83.8 ± 20.7	77.2 ± 15.7	N.S.
KOOS Sports (points)	63.3 ± 31.4	55.0 ± 36.5	N.S.
KOOS QOL (points)	60.4 ± 35.3	67.2 ± 27.7	N.S.
Forgotten Joint Score (points)	52.1 ± 40.1	47.4 ± 40.6	N.S.
KFH (°)	3.1 ± 1.3	7.9 ± 5.3	0.06
Midstance KFA (°)	14.0 ± 4.3	19.2 ± 5.4	0.12
KAA excursion during swing (°)	4.7 ± 1.4	7.9 ± 2.2	0.03
KAM1 (%BW*Ht)	2.21 ± 0.64	2.09 ± 0.31	N.S.
Peak KFM (%BW*Ht)	2.34 ± 1.16	3.78 ± 1.42	0.12
Peak KIRM (%BW*Ht)	0.88 ± 0.14	0.64 ± 0.09	0.02

## Data Availability

All underlying data are in the text, tables and figures.

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
