# Peer review of "Kinematic Comparison between Medially Congruent and Posterior-Stabilized Third-Generation TKA Designs"

_jfmk, 2021, doi:10.3390/jfmk6010027_

Round 1

Reviewer 1 Report

The authors investigated two different TKA implant designs with regards to the in vivo kinematics. The authors conclude that the data reported do not show a relevant difference between the two implant designs. 

Author Response

Dear Reviewer,

thank you for taking the time to review our manuscript. Your comments have been very valuable to us. The manuscript underwent major changes. We modified all sections of the manuscript, according to the Reviewers suggestions.

Specific answers:

1) The English editing of the manuscript has been reviewed by a native English speaker (JLA; lines 359-361)

2)  The Methods section of the manuscript has been modified as suggested. Please, note the following changes: Lines 80-81; Lines 88-90; Lines 144-153;

3) Multiple references have been added (References 6, 7, 8, 14)

4) We agreed with the reviewer: the authors concluded that only few differences between the two designs have been demonstrated (Lines 257-257). The Medial Congruent design (MC) showed more peak internal rotation moment during stance than PS (lines 309-310); there was a significant increase in the knee adduction angle (KAA) during the entire gait analysis in the posterior-stabilized (PS) group when compared to the MC group (lines 301-303).

Thank you again for reviewing our manuscript.

Reviewer 2 Report

The manuscript is a retrospective case–control study aimed to compares knee kinematic in two groups of patients who underwent primary total knee arthroplasty. The authors analyzed two different modern designs: medially congruent (MC) and posterior-stabilized (PS). Finally, ten total knee arthroplasty patients (4 PS, 6 MC) having a successful clinical outcome have been evaluated through 3D knee kinematics analysis performed using a multi-camera optoelectronic system and a force platform.

I read the article with interest, the title is well thought out and faithfully reflects the content of the study. The abstract is adequately developed, and it is useful to frame the characteristics and purpose of the study.

In the introduction, the characteristics of the study have been described. In materials and methods inclusion criteria, exclusion criteria and gait analysis have been adequately developed. The use of figures is adequate for better understanding. The discussion is sufficiently developed.

Nevertheless, some major changes are needed to be considered suitable for publication.

Comment 1: In the introduction: “Many patients still complain of subjective micro- or macro-instability, altered proprioception, abnormal knee joint awareness: their dissatisfaction can be confirmed by of poor range of motion (ROM), chronic effusions, inability to use the stairs in a reciprocating way, inability to kneel or squat and occasional antalgic gait.” Are there any objective parameters evaluated? and with what criteria? Please specify that this is your experience or add a quote about it. For example (Wylde V. et al (2019) "Kneeling ability after total knee replacement").

Comment 2: In materials e method: How many patients participated in this study? How many of these were males and how many females? Are there statistically significant differences between the two sexes?

Comment 3: In materials e method: “All patients received a third generation TKA design (Persona, Zimmer-Biomet, USA) performed by a single surgeon (PFI)” Could you specify the surgeon's experience with this type of surgery and with the prosthesis you have chosen for the study?

Comment 4: In the discussion: "The analysis of the different gait phases in this study showed slight differences when compared to the current literature." Please clarify this statement or add any bibliographic references.

Comment 5: Finally, additional English editing is needed. The Non-Native Speakers of English Editing Certificate was not signed.

Author Response

Dear Reviewer,

thank you for the time you spent reviewing our manuscript. We really appreciated your comments and recommendations. We incorporated many of your suggestions in the revised manuscript that has been resubmitted.

Specific answers:

Comment 1 (Introduction) : Lines 38-42. We added 3 references (Reference 6, 7, 8) which were used to reinforce the concept that many TKA patients are still limited in their daily functions: we focused on instability (reference 6, as suggested by the reviewer) and kneeling ability (reference 7 and 8, as suggested by the reviewer).

Comment 2 (Materials and Methods): Patients demographics are described in Table 1 (please, check lines 86-87; please, check Table 1).

Comment 3 (Materials and Methods). Surgeon's experience has been specified in lines 89-90 (please, check References 21-23 which were added in this paragraph).

Comment 4 (Discussion). We highlighted that "...this study showed slight differences when compared to the current literature..." (Lines 266-268). We added 5 references (line 268; references 32-36).

Comment 5 (English editing):  A native English speaker reviewed the manuscript (JLA; lines 359-361). 

Thank you again for the precious suggestions finalized to improve the quality of our work. 

Round 2

Reviewer 1 Report

The authors have submitted a revised version of their paper. The main flaw (low number of cases) haven't changed. 

Author Response

Dear Reviewer,

thank you for reviewing our manuscript one more time. We really appreciated all suggestions.

Regarding your final comment, we acknowledged in the Discussion section (Lines 342-346) that the small number of participants represents a major limitation of the current study: unfortunately, this study was conducted during the Covid-19 pandemic and patients access to the gait analysis Lab was limited by institutional (Stanford University) and federal (US Department of Veterans Affairs) regulations. The study funding was also limited to the 2019-2020 academic year. We hope that you'll be able to recognize the difficult circumstances that we, as many other researchers worldwide, encountered in these unprecedented times. 

Round 3

Reviewer 1 Report

The reviewer encourages the authors to do the study with a larger number of cases. In its current form, it is impossible to derive any clinical relevance from the data.